

# Erosive phenomena in the Kaulon archaeological site: origins and remedies

Giuseppe Barbaro[1], Salvatore Siviglia[2], Giandomenico Foti[1], Carmelo Luca Sicilia[1], Maria Clorinda Mandaglio[1], Ferdinando Frega[3], Pierfabrizio Puntorieri[1]

[1]DICEAM Department, Mediterranea University of Reggio Calabria, Reggio Calabria loc. Feo di Vito, 89122, Italy
[2]Calabria Basin Authority, Catanzaro Loc. Germaneto, 88100, Italy
[3] Department of Civil Engineering, University of Calabria, Arcavacata di Rende, 87036, Italy

*Correspondence to*: Giuseppe Barbaro (giuseppe.barbaro@unirc.it)

**Abstract.** This paper analyses the erosive phenomena which affected the coast of Monasterace Marina, in the province of

Reggio Calabria (Italy), where the Kaulon archaeological site is located on a dune and which extends along the coast for 1 km. The phenomena which began in the last 50 years have been reinforced in the new millennium. Recently, a storm that occurred in early December 2013 eroded much of the beach and caused destruction of parts of the dune, with sliding of some archaeological finds onto the beach below. Following this storm, the Provincial Administration of Reggio Calabria in January 2014 built a barrier. It was only a temporary intervention however. In fact, in early February 2014 another storm

caused further damage to the beach and to the site. As a result of this, in 2015, the Calabria Basin Authority proposed a design solution which is part of the Master Plan of solutions to mitigate the risk of coastal erosion in Calabria.

In this paper, all the possible causes of erosion are investigated through the analysis of different types of data: bathymetric, cartographic, geological, geomorphological, sedimentological, climatic and wave climatic. This analysis is carried out for different historical periods over the last 50 years. Also analysed were extreme sea storms, coastal and river sediment

transport, land cover and average rainfall, in order to understand the erosive phenomena. Finally, the huge amount of data obtained in this study was used to analyse the project proposal for different solutions to the problem.

## 1 Introduction

Coastal areas are one of the most populated territories of the world. In fact, 37% of the world's population lives within 100 km of a coast, while about 50% live within 200 km (Burbridge, 2004). In particular in Calabria, recent studies have shown

that only 30% of the coastal area is not subject to coastal erosion, a very small percentage.

Given this context, the definition of a consistent strategy to protect the coastal areas is very important, in order to guarantee safe conditions in those areas exposed to both environmental and anthropic actions. Furthermore, we must consider that this phenomenon is mainly due to a deficit of sediments, caused by both natural and anthropogenic factors.

Regarding the natural factors, we are reminded of the subsidence which involves the lowering of the seabed due to soil

compaction, or the raising of the mean sea level produced by climate change.



In addition to these natural causes, unfortunately, are added anthropogenic causes that tend to amplify this phenomenon, making coastal erosion even more destructive. The biggest problem is linked to the reduction of river solid contributions to the sea, which are not enough to ensure the equilibrium of the beaches. In fact in the last few years, we have witnessed the building of dams for hydroelectric purposes, for lamination of floods, or for irrigation, which has blocked the natural sediment transport. To this we must add the indiscriminate excavations of river beds and the building of embankments. Likewise, the construction of massive infrastructures (such as ports, harbours, breakwaters, etc.) and the urbanization of the coast, with the consequent destruction of coastal dunes and the increase in building near the shorelines, has modified or blocked the longshore sediment transport.

This phenomenon can produce environmental disasters, as for example, what happened in Saline Joniche in the province of Reggio Calabria, where hundreds of metres to tens of kilometres of beaches have disappeared which were very important to the local economy (Barbaro, 2013). Furthermore, dunes work like sand reservoirs where the sea can be drawn during exceptional storms, and their absence makes the beach more vulnerable to storm action (Tomasicchio et al., 2011).

To understand and reduce the problem of coastal erosion it is useful to carry out periodic monitoring of the shoreline (through bathymetric surveys of the seabed, topographical surveys of the emerged beach, and size analysis of sediments), and to study the phenomena which are involved in coastal dynamics (waves, currents, longshore and river sediment transport, etc.). Moreover, it is very important to know the past and present management plans, the tourist use of the coast, and any structures built near the shore and their effects on neighbouring coasts.

It is possible to use different design concepts to solve this problem (Salvadori et al., 2013, 2014, 2015). The first option involves the protection of the coast without structures, the so-called "zero option." This option is applicable mainly in the areas in which the environmental aspect must be safeguarded and where carrying out such work would involve a substantial environmental impact, or in where barrier islands, lagoons or estuaries are present. The "one option", instead, consists of the implementation of a soft intervention, called a "nourishment", which involves pouring sand into an area already eroded, in order to partially or totally recreate a pre-existing beach. The sand may be taken from underwater quarries, river mouths or accumulation areas along the coast (for example, in the proximity of a harbour pier). The "two option" instead involves the presence of structures, usually barriers or groynes, and this choice depends on the direction of more intense waves. It is characterized by low costs of the project life, but which need frequent maintenance (the same as for beach nourishment). The "three option" consists of a mix of those actions described above (eg nourishment and works).

In addition to the above-mentioned monitoring, is also necessary to provide proper management over the interventions, which must cover three aspects, the first being the engineering aspect, to protect coastal flooding, the second being the economic, to support productive activities and the third being environmental, to preserve priority habitats.

Furthermore, coastal erosion risk can be estimated through the analysis of a multidisciplinary geo-indicators set, aiming to describe the physical, environmental and socio-economic factors. The index of socio-economic sensitivity was expressed in terms of five indicators: housing, business, tourism, infrastructure and environment. These have been carried out at the



regional and municipal level, using census data of individual sections of the municipalities. The whole area was subdivided in homogeneous sectors, to better estimate the indicator values.

Finally, it should be noted that temporary interventions are often put in place, examining only the beach where the structures will be built, and not taking into account interactions with the surrounding coasts of that physiographic unit. It is important,

however, to change the programme strategy and management planning to take into account all the above factors which influence the equilibrium of coastal areas, and to plan to scale physiographic units (for example, a groyne may intercept a part of the sediments transported by waves and ensure a relative stability in the protected area but, at the same time, it may impoverish other areas behind the structure by shifting the problem of erosion). For this reason the phenomenon should be studied from a holistic point of view to include all the aspects involved (Barbaro et al., 2014a; Barbaro, 2016), in particular

wave and weather climate analysis (Barbaro, 2007; Barbaro, 2011; Boccotti et al., 2011; Arena et al. 2013, a, b, c; Barbaro and Foti, 2013; Barbaro et al., 2013 a, b; Boccotti, 2015), and the contribution of longshore and river sediment transport (Tomasicchio et al., 2007;Tomasicchio et al., 2013; Barbaro et al., 2014b; Mandaglio et al., 2015, 2016 a, b, c; Tomasicchio et al.. 2015).

Recently, the Calabria Basin Authority has developed the "Master Plan of solutions to mitigate the risk of coastal erosion in

Calabria" to define criteria and methods relating to the calculation of the risk of erosion and priority classification of various areas. This tool is necessary to limit and possibly solve problems due to the specific climatic and coastal characteristics of this region, and also to avoid the incorrect design of coastal structures.

This paper describes the erosive phenomena that affected the coast of Monasterace Marina in Italy, where the Kaulon archaeological site is located on a sand dune. In particular, cartography data consists of aero photogrammetry by CASMEZ

(1958), IGM (1985), and aerial photos (1998 and 2008), provided by the Calabria Basin Authority (ABR), and satellite imagery provided by Google Earth Pro. The Wave time series starts on 10/01/1986 and finishes on 31/03/2006 and is provided by the Met Office database. To evaluate changes in river sediment transport, hydraulic structures, land cover data and weather and climate data were analysed, specifically the land cover data used was related to year 2000 and year 2006 (Corine Land Cover project), which is freely available on the government agency website "Istituto Superiore per la

Protezione e la Ricerca Ambientale (ISPRA)", while the weather and climate data are average rainfall that were registered in the station near the area (Serra San Bruno, Monasterace Punta Stilo, Stilo Ferdinandea, Mongiana, Fabrizia, Stignano and Santa Caterina dello Ionio). This analysis was carried out for different historical periods over the last fifty years with the objective of analysing both the causes of coastal erosion and the interactions that exist between them. A temporary and a final solution are also described in the paper.

**2 Main aims**

- ▪ Describe the peculiarities of the Kaulon archaeological site, its climatic and sedimentological features;
- ▪ identify the causes of coastal erosion in the location under examination. The steps are:




- evaluate the historical evolution of the coastline, through the comparison of cartography data, provided by Calabria Basin Authority (which consists of aero photogrammetry by CASMEZ, 1958 and IGM, 1985, aerial photos, 1998 and 2008), and satellite imagery provided by Google Earth Pro;

- evaluate the deep water wave climate, for different historical periods;

- evaluate the sediment contribution, through the estimation of longshore sediment transport, river transport, land cover and average rainfall;

▪ describe the temporary solution used by the Provincial Administration of Reggio Calabria, with the aim of defending the artistic heritage;

▪ describe the final solution, proposed by the Calabria Basin Authority, to resolve the erosion and to defend the coast

of Monasterace.

## 3 Site description

### 3.1 Geographic classification

The site is located near the town of Monasterace Marina (Southern Italy) and between the mouth of the rivers Assi and Stilaro (Fig. 1a). The coastline has an inclination of 15° from the North. The site is affected by the prevalent winds that blow

from the South and South-Easterly and from the North and North-Easterly directions. The most severe storms arise mainly from the South and South-Easterly direction, where the fetch is up to 700 km (Fig. 1b), and they are concentrated in the winter season.

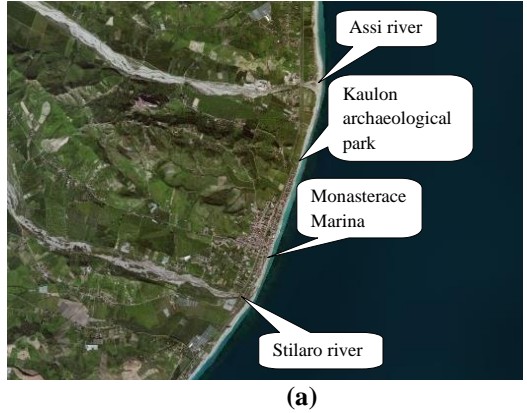
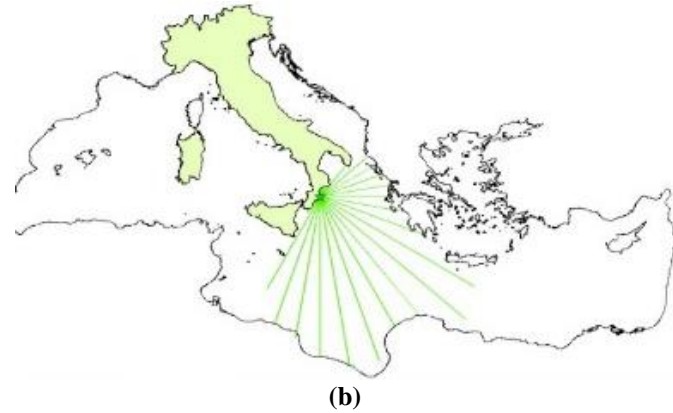

               **(a)**                       **(b)**

**Figure 1.** Geographic Classification. **(a)** Location of the site. **(b)** Fetches.

### 3.2 Geological, geomorphological and sedimentological framework

The area under consideration is situated on the Ionian side of Calabria on the stretch of coastline between the mouths of the Stilaro and the Assi rivers. The zone is located in an intermediate position between the Squillace Gulf to the North and that of Bovalino-Siderno to the South (Fig. 2).



The region is characterised by recent tectonic uplift accompanied by compressive phenomena eastward and distensive and sinking phenomena to the west. The intense and recent tectonic activity has provided a morphology subject to intense erosion and gravitational processes as occurs in many rocky outcrops in other zones of Calabria (Aliperta et al., 2012; Mandaglio et al., 2015, 2016 a, b).

The geological formations constituting the area are sedimentary rocks of Holocene-Recent age overlapped to the oldest sedimentary rocks (Mio-Pliocene-Pleistocene) and to crystalline-metamorphic units of Paleozoic age.

Overall, the litho-stratigraphic sequence, starting from the most ancient rocks, can be summarized as follows: clays and polychrome silts (Miocene Inf.-Med.), clays and whitish marl (Lower-Middle Pliocene), conglomerates and sands (Pliocene Sup. - Calabriano), sandstones and clay silts (Middle - Superior Pliocene), conglomerates and mica sands (Sup. Pliocene-

Calabriano "Ghiaie di Messina"), conglomerates and sands (Pleistocene), debris fan (Holocene), solifluction products (Holocene), stabilised alluvial deposits (Holocene); coastline and mobile alluvial deposits (Holocene), stabilised sand dunes, mobile sand dunes.

From the geomorphological point of view, the area is characterised by a narrow coastal plain and by hills which connect to the reliefs of the Serre. The coast is characterised by low beaches and well developed dune systems and a sea bottom with

moderate slope (2-3%). The coast is located at the outer edge of a thin coastal plain whose natural balance has been affected by intense erosion activity caused by sea waves during storms, and by coastal currents. Moreover, some human activities have acted to divert the solid longshore transport offshore and subtract a considerable amount of shoreline sediments. The effect of these processes was a considerable receding of the coastline, which led to the building of different protective measures inside the physiographic unit with positive mitigation effects.

The coast has a regular morphology and, towards the interior, the landscape changes, first outlining some reliefs with narrow terraced areas and then reliefs with steep slopes, segmented by numerous and deep valleys. The hilly area shows a dorsal structure with valleys; the reliefs are mainly constituted of clay with marl and sands and conglomerates, often deeply degraded (Gullà et al., 2006; Mandaglio et al., 2016c). The slopes are locally intersected by gullies and at the foot of the slopes there are alluvial fans coming from the intense solid transport sediments of the rivers (Barbaro et al., 2012); the

smaller valleys have a "U" shape with a flat valley floor with alluvial deposits.

The hydrography is characterised by a series of rivers (*fiumare*) with dendritic and fin patterns, with high drainage density, perpendicular to the coastline with a typical braided course.

From the sedimentological and grain size distribution point of view, the beach sediments are composed of sand and light gray gravels, with $D_{50}$ equal to 5 mm next to the isobaths of + 1.0m, $D_{50} = 0.7$ mm next to the isobaths -3.0m and $D_{50}$ equal

to 0.82 mm at the isobaths -7.0m.





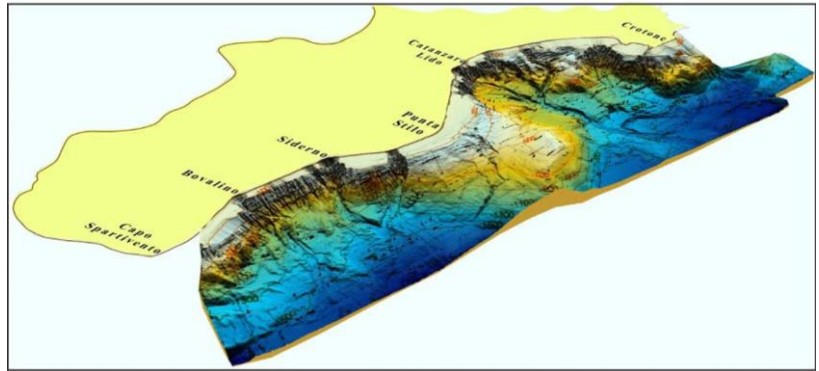

**Figure 2.** Coastline and bathymetry.

### 3.3 The Kaulon archaeological park

Kaulon was a colony of Magna Grecia, founded in the eighth century BC and discovered in 1911 by the archaeologist Paolo
Orsi. In the heyday, dating from the fourth century BC, it extended over an area of 47 hectares and had a population of
10,000 inhabitants. It is one of the richest Magna Greek cities and was in fact the first to mint silver coins.

Kaulon consisted of a major urban centre, surrounded by walls and placed at sea level, organized under the regular system
Hippodamian (named after the architect Hippodamus of Miletus): this system was characterized by a narrow road network,
arranged upstream to downstream to facilitate the flow of water, which intersected orthogonally wide roads, denominated
"plateiai", and forming blocks divided into lots, each of which was divided into two quadrangular houses of 17m for 17m in
size; the road was made from sand and gravel, with the addition of ceramic fragments.

In the southern part of the site there was a Doric temple (Fig. 3 a), probably dedicated to Jupiter Homarios, of which only the
base, the altar, the steps and other sacred structures are now visible. The area in front of the temple, currently covered by the
sea, was occupied by the town, as evidenced by archaeological finds which bear witness to the gradual erosion of the
coastline.

Besides the Doric temple, the site also has the Pillbox (Fig. 3 b), a thermal building which was later transformed into a place
of worship, in whose interior was recently discovered (in 2012) a 25 $m^2$ mosaic floor depicting dolphins and dragons, dating
back to the end of the IV and the beginning of the III century BC, which is among the largest of the Magna Greece. There is
the House of the Dragon (also known as the House of Grotesque Character), discovered in the 1960s and dating from the
third century BC, which belonged to a wealthy family and was turned into a residential sector and a sector of representation
which takes its name from a mosaic depicting a sea dragon (called Drakon, Fig. 4), whose back is covered with spines and
which has the tail of a fish, which is among the most ancient of Calabrian mosaics.





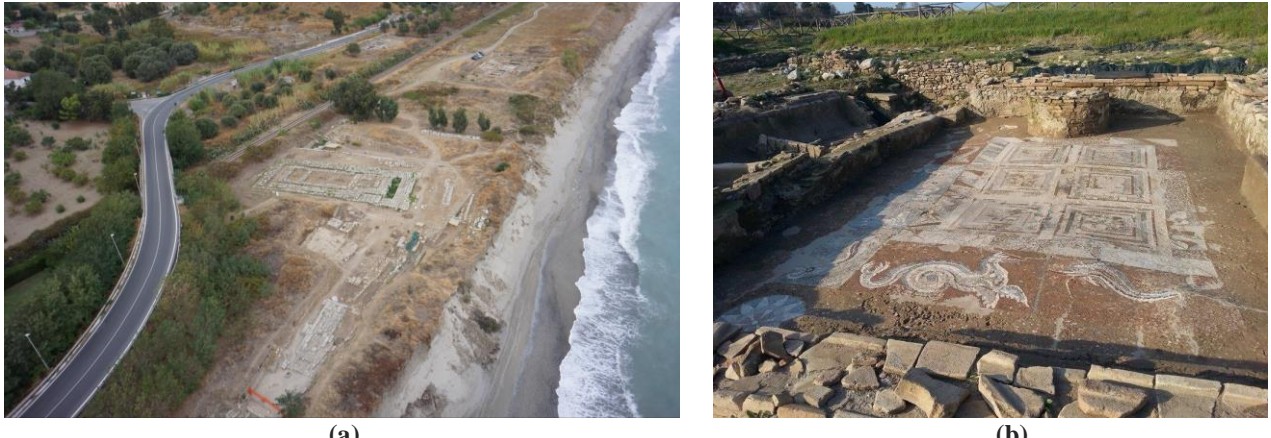

(a)      (b)

**Figure 3. (a)** View of the Kaulon archaeological site. **(b)** The Pillbox and its mosaic.

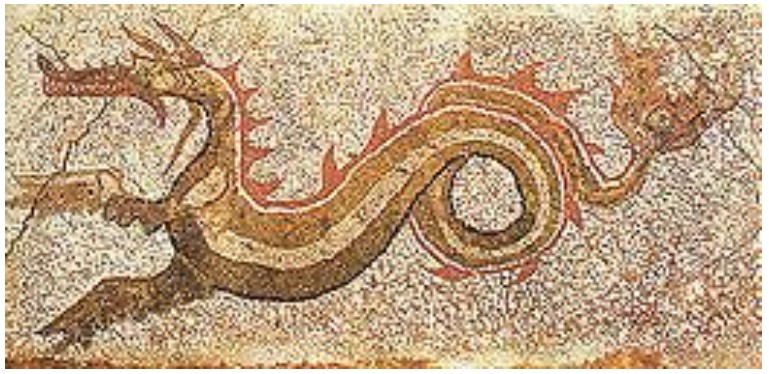

**Figure 4.** The Drakon mosaic.

## 4 Historical evolution of the coastline

5    Cartographic data was processed using QGS 2.8.3, the available data consisting of aero photogrammetry by CASMEZ (1958), IGM (1958) and aerial photos (1998 and 2008). All data was provided by the ABR, Calabria Basin authority and Google Earth Pro (Fig. 5 and Table 1).

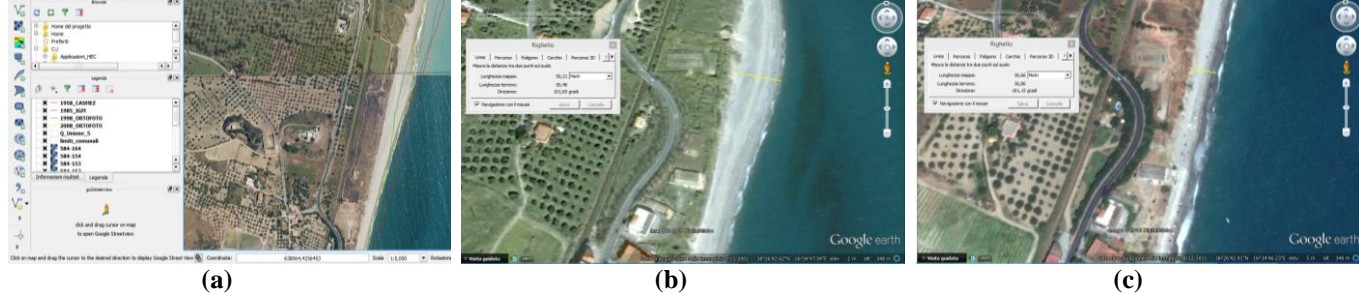

(a)      (b)      (c)

**Figure 5. (a)** Coastline evolution between 1958 and 2008 (from ABR). **(b)** Coastline at 13/03/2001 (from Google Earth Pro). **(c)** Coastline at 13/08/2011 (from Google Earth Pro).





**Table 1.** Coastline comparison between different periods from 1958 to 2011 (red: erosion, green: accretion, white: equilibrium).

| Data | Beach Width [m] | Variation [m] | Variation [%] |
|------|-----------------|---------------|---------------|
| 1958 | 57 | - | - |
| 1985 | 39 | -18 | -32 |
| 1998 | 47 | +8 | +21 |
| 13/3/2001 | 50 | +3 | +6 |
| 2008 | 29 | -21 | -42 |
| 13/5/2010 | 30 | +1 | +3 |
| 11/3/2011 | 30 | - | 0 |
| 12/8/2011 | 30 | - | 0 |

The erosion in front of the archaeological site is evident, as it has developed over the last 50 years in two phases. Due to the lack of continuous monitoring of the coastline, the following events happened in the time gaps. The first event happened between 1958 and 1985, during which phase more than the 30% of the original beach was eroded. The second time gap phase was between 2001 and 2008, during which phase the erosion was more evident, and in fact was over 40%. Between these two phases another one is evident, characterized by beach accretion between 1985 and 1998. Finally, between 2008 and 2011 the coastline appears to have been in an equilibrium state.

## 5 Wave climate

The wave climate has been investigated via the ABRC-MaCRO software, developed by HR Wallingford Ltd. This software allows us to obtain time histories of wave data, starting from the information available at the Met Office database. This database is composed of data reconstructed via the European Wave Model starting from wind field data. The software is based on the HINDWAVE model (Ewing 1989), which is implemented starting with the following input data: geometric characteristics of the area under investigation and wind velocity field in that area. The calculation is carried out in two stages: first, a table with all the combinations of wave data compatible with the characteristics of the site is defined, then wind records are analysed to identify which wave conditions are better correlated with current records.

The model has been calibrated via the buoy wave data recorded in Crotone (Ionian Sea) and Cetraro (Tirrenian Sea). This data is provided by the Rete Ondametrica Nazionale (RON).

The time series obtained from the software starts on 10/01/1986 and finishes on 31/03/2006. The whole sea states have been divided into different significant wave heights classes and into different sectors. The entire data has been also subdivided in two irregular sets of time periods, in accordance with the coastline evolution (1986-2001 and 2001-2006) (Table 2). Starting from the time series, the following were calculated: frequency, significant wave height, peak period, energy flux, probability of exceedance, directional probability of exceedance, return period, directional return period, mean persistence, run-up and mean persistence of the run-up.

**Table 2.** Number of sea states, in ordered of significant wave height and time periods.




| Time periods | 1986-2006 | 1986-2001 | 2001-2006 |
|---|---|---|---|
| Hs [m] | Number of sea states | | |
| 0.0-0.5 | 84114 | 60326 | 23788 |
| 0.5-1.0 | 47709 | 36527 | 11182 |
| 1.0-1.5 | 17152 | 13315 | 3837 |
| 1.5-2.0 | 4269 | 3316 | 953 |
| 2.0-2.5 | 2248 | 1669 | 579 |
| 2.5-3.0 | 2135 | 1810 | 325 |
| 3.0-3.5 | 1179 | 1028 | 151 |
| 3.5-4.0 | 94 | 76 | 18 |
| 4.0-4.5 | 288 | 258 | 30 |
| 4.5-5.0 | 69 | 69 | 0 |
| 5.0-5.5 | 21 | 21 | 0 |
| 5.5-6.0 | 20 | 20 | 0 |
| Total | 159298 | 118435 | 40863 |

## 5.1 Frequency, significant wave height, peak period and energy flux

The frequency of occurrence (Fig. 6) is given by the equation:

$$f(\Delta\theta) \sum_{i=1}^{n} f(\Delta H_i, \Delta\theta) \,, \tag{1}$$

where n is the number of classes of significant wave height, $\Delta H$ the extent of a class of significant wave height, and $\Delta\theta$ the

5    extent of a wave sector, also:

$$f(\Delta H, \Delta\theta) = \frac{N\,(\Delta H, \Delta\theta)}{N_{tot}} \,, \tag{2}$$

with N ($\Delta H$, $\Delta\theta$ ) the number of sea states belonging to a class $\Delta H$ and to a sector $\Delta\theta$ and $N_{tot}$ total number of sea states.

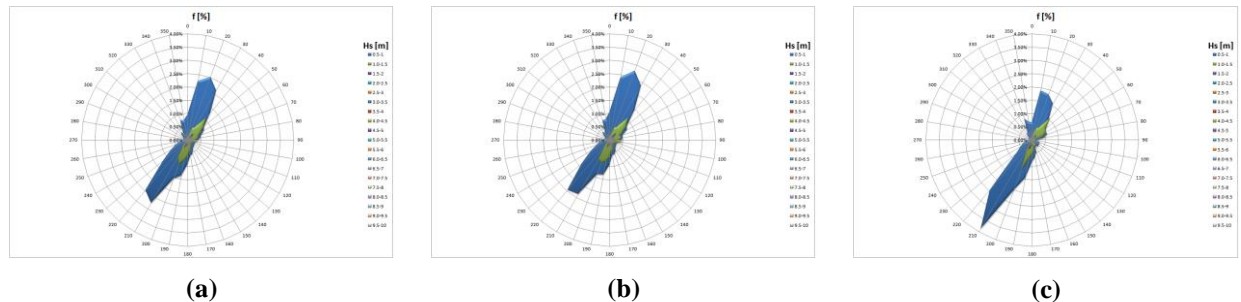

(a)                                  (b)                                  (c)

**Figure 6.** Frequency of occurrence. **(a)** 1986-2006. **(b)** 1986-2001. **(c)** 2001-2006.

The mean significant wave height (Fig. 7) is given by the equation:

10    $$\overline{H_{s_0}}(\Delta\theta) = \frac{\sum_{i=1}^{n} \overline{H_s}(\Delta H_i, \Delta\theta)}{N_{tot}(\Delta\theta)} \,, \tag{3}$$




with $\overline{H_{s_0}}(\Delta H_i)$ the mean significant wave height of ΔH class and $N_{tot}$ (Δθ) the total number of sea states of Δθ sector.

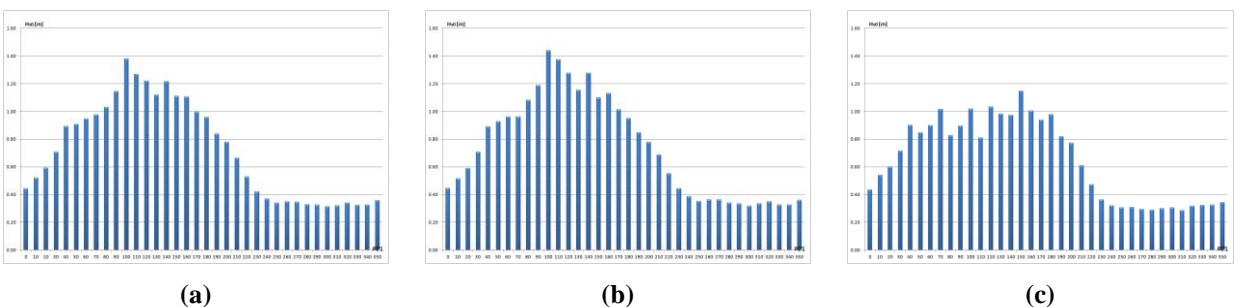

**Figure 7.** Mean significant wave height. **(a)** 1986-2006. **(b)** 1986-2001. **(c)** 2001-2006.

The peak period (Fig. 8) is given by the equation:

$$\overline{T_p}(\Delta\theta) = \frac{\sum_{i=1}^{n}\overline{T_p}(\Delta H_i, \Delta\theta)N(\Delta H_i, \Delta\theta)}{N_{tot}(\Delta\theta)} , \qquad (4)$$

5   with $T_p$ (ΔH$_i$, Δθ) the peak period belonging to a class ΔH and to a sector Δθ.

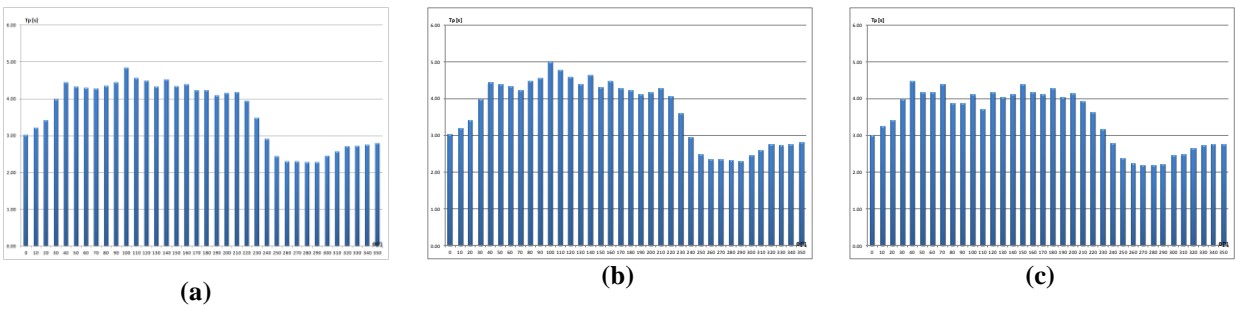

**Figure 8.** Peak period. **(a)** 1986-2006. **(b)** 1986-2001. **(c)** 2001-2006.

The mean energy flux (Fig. 9) is given by the equation:

$$\Phi(\Delta\theta) = \sum_{i=1}^{n}\phi(\Delta H_i, \Delta\theta) , \qquad (5)$$

with:

10   $$\Phi(\Delta H, \Delta\theta) = 986.5\ \overline{H_s^2}(\Delta H)T_p(\Delta H, \Delta\theta)\ f(\Delta H, \Delta\theta) , \qquad (6)$$





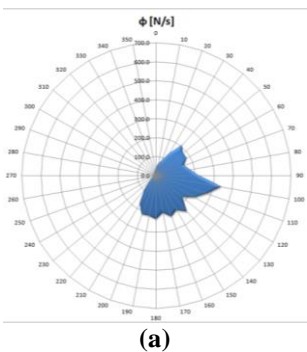
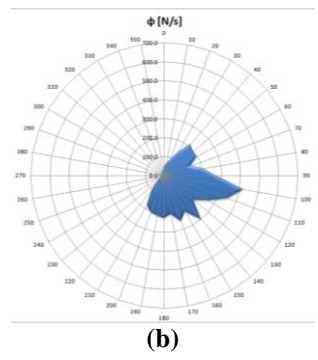
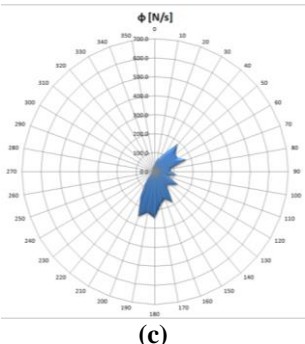

| (a) | (b) | (c) |

**Figure 9.** Mean energy flux. **(a)** 1986-2006. **(b)** 1986-2001. **(c)** 2001-2006.

The data across the whole time period shows that the coast front Kaulon has been affected by frequent sea states from the northeast and southwest, with a maximum significant wave height about 1.4m and a maximum peak period about 5 s, both from the sector centred on the 100° N direction. Furthermore, the greatest part of the wave energy is between 40° and 200° N direction, with a maximum value centred on the 100° N direction.

The data referred to in the periods before and after 2001 shows that before this time the significant wave height was below the threshold of 5.5-6.0 m, and after 2001 it was down to 4.5 m. The results obtained with the time series before 2001 are very similar to the entire time series while those obtained after 2001 are very different. In the latter case, in fact, the sea states from the southwest are more frequent than those from northeast, the maximum value of significant wave height being about 1.2 m from the sector centred on the 150° N direction and the maximum value of peak period being about 4.5 s from different directions. Moreover, after 2001 there was no energy peak in the 100° N direction as it was in the 200° N direction. Comparing the time evolution of the coastline with the time evolution of the wave climate it can be observed that the beach accretion occurred when the wave climate was more severe (1985-2001), and, on the contrary, the erosion occurred when the wave climate was less severe (2001-2006).

## 5.2 Probability of exceedance, return period and mean persistence

The probability of exceedance (Boccotti, 1986) is given by the Weibull distribution as follows:

$$P(H_s > h) = \exp\left[-\left(\frac{h}{w}\right)^u\right] \tag{7}$$

u and w being parameters associated with a certain location; from a physical perspective w is a scale factor, while u provides information on data at various probability thresholds. In addition, the following directional probability of exceedance is employed (Arena and Barbaro, 1999):

$$P(H_s > h; \theta_1 < \theta < \theta_2) = \exp\left[-\left(\frac{h}{w_a}\right)^u\right] - \exp\left[-\left(\frac{h}{w_\beta}\right)^u\right], \tag{8}$$





where $w_\alpha$ and $w_\beta$ are parameters associated with a certain section of wave propagation ($\theta_1$, $\theta_2$). Eq. 7 is useful to obtain a probabilistic description of the sea states belonging to a certain interval of wave propagation. By distinguishing different sectors of wave propagation, a dominant sector can be identified. That is, the one propagating the most relevant fraction of wave energy over one given year. By analysing Eq. 8, it can be seen that this is the one with the largest $w_\alpha$ value (if there are

similar $w_\alpha$ values, then the main sector of wave propagation relates to the smaller $w_\beta$ values).

The return period of significant wave height values is determined by the closed-form solutions derived via the Equivalent Triangular Storm method proposed by Boccotti (2000) and further developed by Arena et al. (2013a). This method was developed with the objective of providing a simple model for describing storm time-variations. Indeed, this method replaces all recorded storms with triangular shaped storms with storm peak a  and duration b, and assumes the maximum expected

wave heights of the real and triangular storms are equal to each other. A crucial characteristic of this method is the possibility of providing closed form solutions to calculate the relevant return period values. Specifically, the return period of a sea state with significant wave height above a certain threshold (Table 3 and Fig. 10) is given by the following equation (Boccotti, 1986):

$$R(H_s > h) = \frac{\overline{b}(h)}{1+u\left(\frac{h}{w}\right)^u} \exp\left[\left(\frac{h}{w}\right)^u\right], \tag{9}$$

$$\overline{b}(h) = b_{10}\left(1.12 - 0.12\frac{h}{a_{10}}\right), \tag{10}$$

where $\overline{b}(h)$ is the regression bases-heights, $a_{10}$ and $b_{10}$ are location dependent parameters determined by a least square minimisation procedure on the calculated a and b parameters. In particular, $a_{10}$ is the average value of the heights of the $10n_{years}$ more severe storms recorded in $n_{years}$ on the fixed location, and $b_{10}$ is the average value of the bases of the $10n_{years}$ more severe storms recorded in $n_{years}$ on the fixed location. For this case, the following values are used: $a_{10} = 3.3$ m; $b_{10} = 82$

hours (Boccotti, 2000, 2014).

The concept of directional return period was developed by Arena et al. (2013a). Specifically, they derived the return period of a sea state with significant wave height larger than a certain threshold and propagating to a certain direction of wave propagation (Table 4) as follows:

$$R(H_s > h; \Delta\theta) = \frac{\overline{b}(h)}{\left[1+u\left(\frac{h}{w_\alpha}\right)^u\right]\exp\left[-\left(\frac{h}{w_\alpha}\right)^u\right]-\left[1+u\left(\frac{h}{w_\beta}\right)^u\right]\exp\left[-\left(\frac{h}{w_\beta}\right)^u\right]}, \tag{11}$$

These quantities were also employed to estimate the mean persistence. That is, the average time during which the significant wave height is above a certain threshold (Fig. 10). It is related to the return period and to the probability of exceedance by the expression:

$$\overline{D}(h) = R(H_s > h)P(H_s > h), \tag{12}$$



**Table 3.** Probability of exceedance parameters and significant wave height threshold for different return periods.

| Period | 1986-2006 | 1986-2001 | 2001-2006 |
|---|---|---|---|
| u | 0.941 | 0.936 | 0.946 |
| w [m] | 0.558 | 0.575 | 0.497 |
| R [years] | | h (R) [m] | |
| 5 | 5.50 | 5.73 | 4.82 |
| 10 | 6.04 | 6.29 | 5.29 |
| 15 | 6.35 | 6.63 | 5.56 |
| 20 | 6.58 | 6.86 | 5.76 |
| 50 | 7.29 | 7.61 | 6.38 |
| 100 | 7.82 | 8.17 | 6.84 |
| 200 | 8.36 | 8.74 | 7.31 |
| 500 | 9.07 | 9.48 | 7.93 |

**Table 4.** Directional probability of exceedance parameters and significant wave height threshold for different return periods in dominant sector.

| Sector [°] | 95-105 |
|---|---|
| $w_\alpha$ [m] | 0.495 |
| $w_\beta$ [m] | 0.479 |
| R [years] | h (R) [m] |
| 5 | 3.64 |
| 10 | 4.22 |
| 15 | 4.55 |
| 20 | 4.78 |
| 50 | 5.50 |
| 100 | 6.03 |
| 200 | 6.56 |
| 500 | 7.24 |





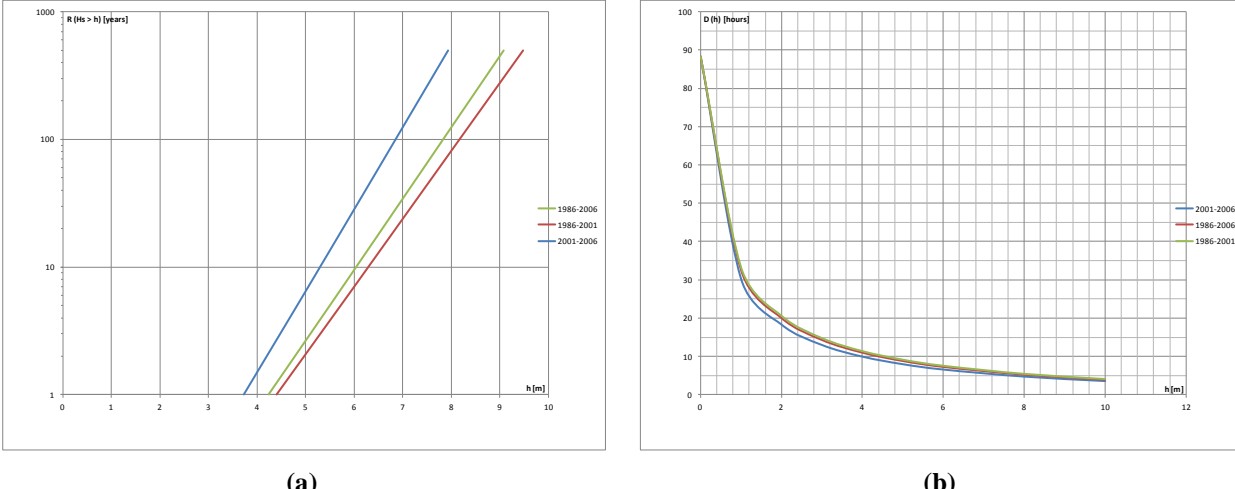

(a)                                             (b)

**Figure 10. (a)** Return period for different time interval. **(b)** Mean persistence.

The analysis of the parameters of the probability of exceedance is in agreement with the results shown in the previous paragraph: in fact, the variations of the parameter u to the different time scales are negligible while the parameter w (scale factor) assumes similar values between the whole time series and the interval 1986-2001 and it is below the previous values in the 2001-2006 interval. A Similar situation is observed for return periods. Finally, the analysis of directional parameters confirms the results obtained from the study of the energy flow in terms of the dominant sector.

**5.3 Longshore sediment transport**

Longshore sediment transport has been evaluated by the Tomasicchio et al. (2013) model. This model is based on the assumption that movement statistics are affected by obliquity only through an appropriate mobility index and that the units move during up- and down-rush with the same obliquity of breaking and reflected waves at the breaker depth (Lamberti and Tomasicchio, 1997; Tomasicchio et al., 1994). A particle will pass through a certain control section in a small time interval $\Delta t$ if and only if it is removed from an updrift area of extension equal to the longitudinal component of the displacement length, $l_d \sin \theta_d$, where $l_d$ is the displacement length and $\theta_d$ is its obliquity.

The longshore sediment transport rate (Fig. 11) can be also expressed as in the following:

$$Q_{LT} = \frac{S_N D_{n50}^3}{T_m} ,$$ (13)

To derive this formula it is assumed that the displacement obliquity is equal to the characteristic wave obliquity at breaking and that a number $N_{od}$ of particles removed from a $D_{n50}$ wide strip moves under the action of 1,000 waves, then the number of units passing a given control section in one wave is:

$$S_N = \frac{l_d}{D_{n50}} \frac{N_{od}}{1000} \sin \theta_{k,b} ,$$ (14)





with:

$$l_d = \frac{1.4\, N_s^{**} - 1.3}{\tanh^2(k\,d)}\, D_{n50} \; , \tag{15}$$

$$N_{od} = \begin{cases} 20\, N_s^{**}(N_s^{**} - 2)^2 & \text{per } N_s^{**} \le 23 \\ \exp[2.72 \ln(N_s^{**}) + 1.12] & \text{per } N_s^{**} > 23 \end{cases} \tag{16}$$

$\quad \sin\theta_{k,b} = \frac{c_{k,b}}{c} \sin\theta_0 \; , \tag{17}$

$$N_s^{**} \cong \frac{0.89\, H_{k,b}}{C_k\, \Delta\, D_{n50}} \; , \tag{18}$$

$$\gamma_b = \frac{H_{s_b}}{d_b} \; , \tag{20}$$

$$c_k = \frac{H_k}{H_s} \; , \tag{21}$$

$$\Delta = \frac{\rho_s - \rho}{\rho} \; , \tag{22}$$

$\quad c_{k,b} = \sqrt{\frac{g\, H_{k,b}}{\gamma_b}} \; , \tag{23}$

Where $l_d$ is the displacement length, $N_s^{**}$ is the modified stability number (Lamberti and Tomasicchio, 1997), $H_{k,b}$ is the characteristic wave height at breaking, $\gamma_b$ is the breaker index, $\Delta$ is the relative mass density of the unit, $c_{k,b}$ is the characteristic celerity at breaking, c is the celerity, $c_{g,b}$ is the wave group celerity at breaking, k is the wave number, $\theta$ is the offshore wave obliquity, $H_k$ is the characteristic wave height and is equal to 1.55 $H_s$ if adopted $H_{1/50}$, otherwise is equal to

1.4 $H_s$ if adopted $H_{2\%}$ (Lamberti and Tomasicchio, 1997)

The entire data has been also subdivided into two irregular sets of time periods, in accordance with the coastline evolution (1986-2001 and 2001-2006). Longshore sediment transport (Fig. 11) is directed by South from North direction in all the intervals considered. The entire interval is about 100,000 m³/year (99,267 m³/year to be exact), in the period 1986-2001 it is slightly greater than that of the entire period (exactly 111775 m³/year) while the period 2001-2006 it is significantly smaller

(62,914 m³/year to be exact).

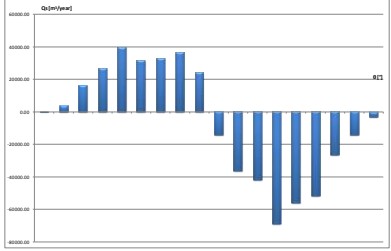
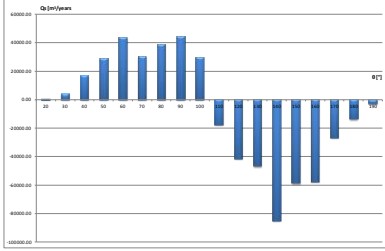
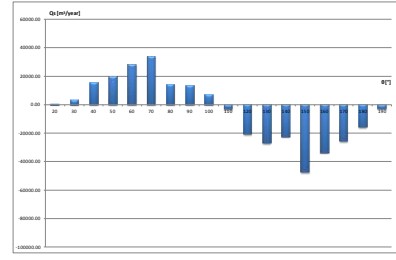


**(a)**          **(b)**          **(c)**

**Figure 11.** Longshore sediment transport. **(a)** 1986-2006. **(b)** 1986-2001. **(c)** 2001-2006.

## 5.4 Run-up

The run-up has been calculated using a probabilistic based on the ETS model in conjunction with the empirical parameterization proposed by Stockdon et al. (2006) for run-up estimation. They proposed the following formula for run-up determination:

$$\frac{R_{u2\%}}{H_0} = K = 1.1 \left\{ 0.35 \, \beta_f \left(\frac{L_0}{H_0}\right)^{1/2} + \frac{1}{2}\left[\frac{L_0}{H_0}\left(0.563 \, \beta_f^2 + 0.004\right)\right]^{1/2} \right\}, \tag{24}$$

Where $R_{u2\%}$ is the 2% exceedance value of run-up, $\beta_f$ is the beach slope, $H_0$ is the wave height at deep water and $L_0$ is the wave length at deep water.

Since random waves are under examination, $H_0$ and $L_0$ will be considered as the significant wave height and dominant wavelength of a sea state.

Return period of a run-up level higher than a fixed threshold is determined as the ratio between a long time interval $\tau$ and the number $N(\tau)$ of run-up levels over the threshold during $\tau$:

$$R(R_{u2\%} > X) = \frac{\tau}{N(\tau)}, \tag{25}$$

where:

$$N(\tau) = \sum_{i=1}^{N} \frac{\tau}{R(H_s > h; \theta_i - \Delta\theta/2 < \theta < \theta_i + \Delta\theta/2)}, \tag{26}$$

and $R(H_s > h; \theta_i - \Delta\theta/2 < \theta < \theta_i + \Delta\theta/2)$ is given by the equation 11.

The mean persistence of the run-up above the fixed threshold is given by the ratio between directional probability of exceedance (8) and directional return period (11).

**Table 5.** Run-up level of fixed return period and relative mean persistence.

| R ($R_u > X$) [years] | $R_{u2\%}$ [m] | D (X) [hours] |
|---|---|---|
| 5 | 0.946 | 8.96 |
| 10 | 1.040 | 8.13 |
| 15 | 1.094 | 7.71 |
| 20 | 1.132 | 7.44 |
| 50 | 1.254 | 6.67 |
| 100 | 1.346 | 6.17 |
| 200 | 1.437 | 5.74 |
| 500 | 1.557 | 5.24 |





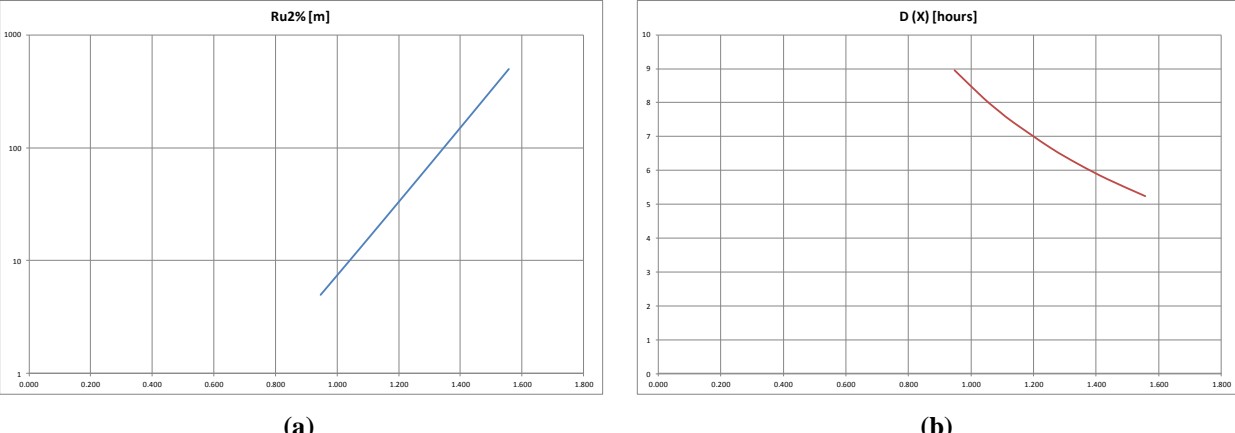

**(a)**                                                  **(b)**

**Figure 12. (a)** Run-up level of fixed return period. **(b)** Mean persistence of run-up level.

The analysis of the results reported in Table. 5 and Fig. 12a, b shows that the run-up values are relatively low (slightly exceeding 1.5 m for a return period of 500 years) but they can still affect the stability of the dune (Fig. 13).

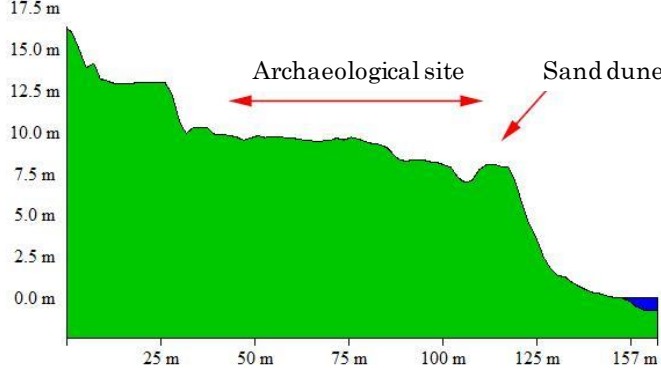

**Figure 13.** Elevation profile of the dune at the Doric temple.

# 6 River sediment transport changes

To evaluate changes in river sediment transport hydraulic structures, land cover data, weather and climate data and river sediment contribution were analyzed

10 ## 6.1 Hydraulic structures

To analyse the hydraulic structures, in the absence of a register of structures, satellite imagery provided by Google Earth Pro were analysed and Hydraulic cards Report was consulted, which is compiled by the Hydrographic Service Monitoring, and provided by the Calabria Basin Authority.




Along the Assi and Stilaro rivers there are few check embankments, and the river beds are formed principally by alluvial deposits.

## 6.2 Land cover data

The land cover data used was related to year 2000 and year 2006 (Corine Land Cover project, fourth level), which was freely

available on the government agency website "Istituto Superiore per la Protezione e la Ricerca Ambientale (ISPRA)". Comparing these two periods it can be said that only 5 km$^2$ (corresponding to 2.5% of the entire surface) has changed (Fig. 14 and Table 6).

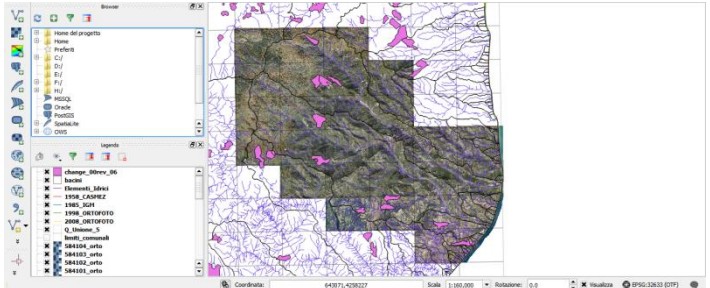

**Figure 14.** Comparison between Corine Land Cover 2000 and 2006 (violet colour) (from GeoPortale ISPRA).

**Table 6.** Changes in use of the soil between 2000 and 2006. Legend: 222: orchards and small fruit, 241: temporary crops associated with permanent crops, 243: crops with the presence of important natural areas, 311: deciduous, 321: natural pastures and meadows, 324: wooded areas and shrub vegetation evolving, 334: areas covered by fires.

| Changes | Area [km$^2$] |
|---------|---------------|
| 222-241 | 0.39 |
| 241-243 | 0.47 |
| 311-324 | 2.20 |
| 321-311 | 0.33 |
| 324-311 | 0.46 |
| 334-311 | 0.25 |
| 334-324 | 0.84 |

## 6.3 Rainfall and temperature time series

Analysed time series were recorded in the gauges near the area, in particular Serra San Bruno, Monasterace Punta Stilo, Stilo

Ferdinandea, Mongiana and Fabrizia (Fig. 15). The time periods are the same as the ones described previously. There are also Stignano and Santa Caterina dello Ionio gauges but these were not considered as they are recently installed and do not have data for all periods examined (Table 7). The area of influence of each gauge was evaluated by the Thiessen polygons



method. Applying this method it is possible to observe that the Mongiana and Fabrizia gauges have no influence on the river

basins in question while the Serra San Bruno gauge has only a small influence on the Assi River basin.

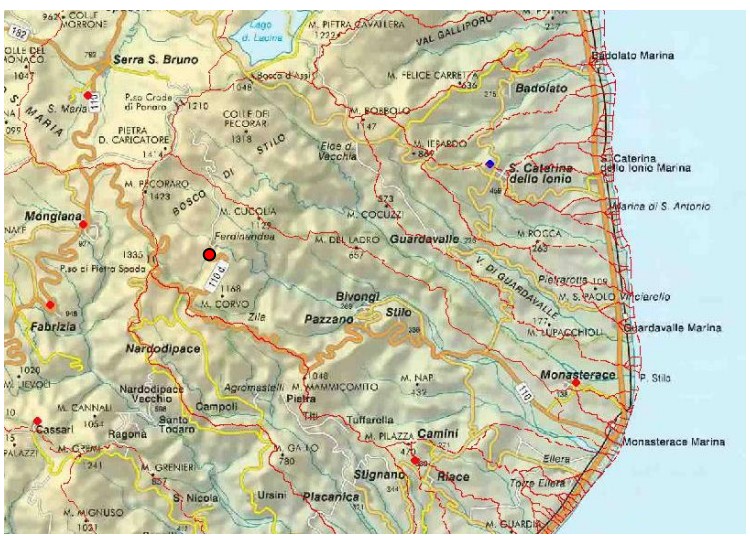

5  **Figure 15.** Assi and Stilaro river basins and nearby gauges.

**Table 7.** Period of registration of each gauge.

| Gauge | Height [m] | Rainfall | | | Temperature | |
|---|---|---|---|---|---|---|
| | | Period | Number of years available | | Period | Number of years available |
| Serra San Bruno | 790 | 1920 - 2016 | 97 | | 1926 - 2016 | 90 |
| Monasterace Punta Stilo | 70 | 1920 - 2016 | 93 | | 1989 - 2016 | 28 |
| Stilo Ferdinandea | 1050 | 1950 - 2005 | 39 | | 1958 - 2005 | 30 |
| Mongiana | 921 | 1928 - 2016 | 48 | | 1992 - 2016 | 25 |
| Fabrizia | 948 | 1920 - 2016 | 88 | | 1992 - 2016 | 25 |
| Stignano | 250 | 2005 - 2016 | 12 | | 2004 - 2016 | 13 |
| Santa Caterina dello Ionio | 459 | 2001 - 2016 | 16 | | Not available | |

**Table 8.** Average annual rainfall of each gauge.

| Gauge | $h_{med}$ (1986-2006) [mm] | $h_{med}$ (1986-2000) [mm] | Variation [%] | $h_{med}$ (2001-2006) [mm] | Variation [%] |
|---|---|---|---|---|---|
| Serra San Bruno | 1615.1 | 1617.9 | 0.2 | 1608.8 | -0.4 |
| Monasterace Punta Stilo | 569.6 | 467.8 | -21.8 | 739.1 | 29.8 |
| Stilo Ferdinandea | 1685.0 | 1710.8 | 1.5 | 1652.7 | -1.9 |

**Table 9.** Average annual temperature of each gauge.

| Station | $t_{med}$ (1986-2006) [°C] | $t_{med}$ (1986-2000) [°C] | Variation [%] | $t_{med}$ (2001-2006) [°C] | Variation [%] |
|---|---|---|---|---|---|
| Serra San Bruno | 10.6 | 10.6 | 0 | 10.6 | 0 |





| Monasterace Punta Stilo | 18.7 | 18.7 | 0 | 18.6 | -0.5 |
| Stilo Ferdinandea | 11.4 | 11.2 | -1.8 | 11.7 | 2.6 |

Through analysis of the results reported in Tables 8 and 9, it was noted that the average temperatures underwent negligible changes in all the time periods examined, while the average rainfall varied significantly only in the Monasterace Punta Stilo gauge, an increase of about 30% in the period 2001 – 2006 (compared to the period 1986 – 2006) and a decrease of about 20% in the period 1986 – 2001 (compared to the period 1986 – 2006).

**6.4 River sediment contribution**

River sediment contribution has been evaluated by the Gavrilovic (1959) model, which is based on an analytical equation to determine the annual volume of detached soil due to surface erosion:

$$Q_t = T\,h\,\pi\sqrt{Z^3}\,A_b\,,\tag{27}$$

Where h is the average yearly precipitation, $A_b$ is the drainage area, T is the temperature coefficient and Z is the erosion
coefficient. The expressions for T and for Z are as follows:

$$T = \left(\frac{t'}{10} + 0.1\right)^{0.5},\tag{28}$$

$$Z = X\,Y\,(G + I^{0.5})\,,\tag{29}$$

Where t' is the average yearly temperature, X is the soil protection coefficient (shown as a function of type of vegetation cover), Y is the erodibility coefficient (shown as a function of type of rock), G is the erosion and stream network
development coefficient (shown as a function of type of the basin erosion) and I is the average slope of the basin. Gavrilovic (1959) suggested a division of the whole basin into different sub areas, hydrographic units, in order to calculate for each hydrographic unit the amount of soil eroded, and to sum up the results in order to obtain a more reliable result.

Gavrilovic (1959) noted that the average erosion rate measured is lower than the calculated one and suggested the use of a reduction coefficient, called the delivery coefficient, which allows us to estimate the amount of eroded sediment which is
deposited along the basin as follows:

$$R = \frac{\sqrt{O\,D}}{0.25\,(L+10)}\,,\tag{30}$$

where O is the perimeter of the basin, D is the average height of the basin and L is the length of the main stream of the basin. Zemljic (1971) proposed another expression to evaluate the delivery coefficient, which is the following:

$$R = \frac{\sqrt{O\,D}\,L_n}{A_b\,(L+10)}\,,\tag{31}$$

where $L_n$ is the length of the river network.



The study was limited to the Stilaro and Assi rivers (Fig. 16 and Table 10), other rivers of the study area being of e negligible size when compared to them.

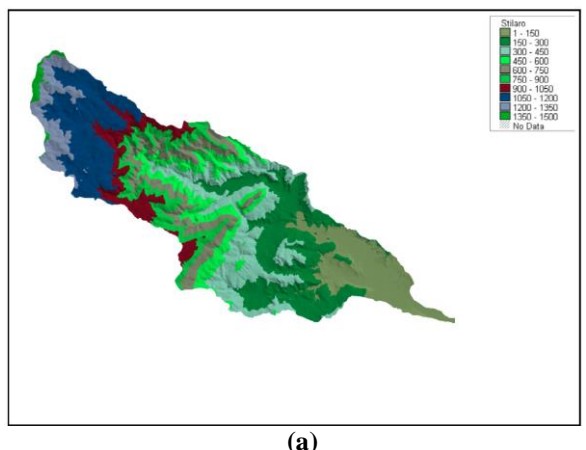
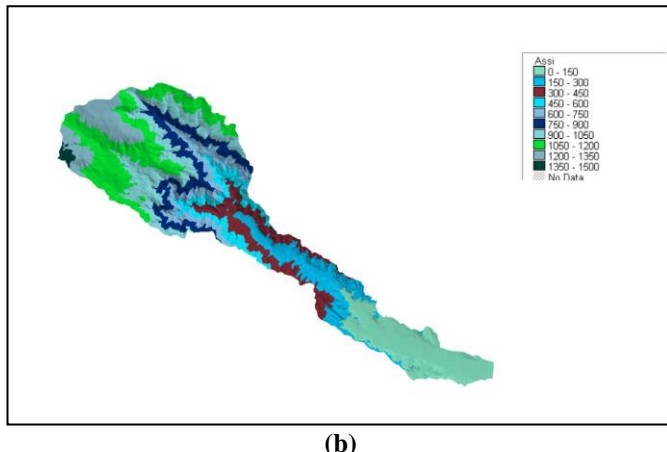

| (a) | (b) |
|---|---|

**Figure 16.** Surface changes to vary the height. **(a)** Stilaro River. **(b)** Assi River.

**Table 10.** Morphological parameters for each river.

| Parameter | Stilaro River | Assi River |
|---|---|---|
| Area [Km$^2$] | 95.04 | 66.50 |
| Perimeter [Km] | 59.17 | 56.77 |
| Main stream length [Km] | 28 | 25 |
| Total stream length [Km] | 462 | 358 |
| Maximum height [m] | 1407 | 1417 |
| Average height [m] | 602.8 | 677.2 |
| Average slope [%] | 33.17 | 34.17 |
| Horton Order | 6 | 6 |
| Gravelius Index | 1.73 | 2.01 |
| Hypsometric integral | 0.41 | 0.48 |
| Run-off time (Giandotti 1934) [hour] | 4.26 | 3.37 |

5   The river sediment contribution is calculated using the Zemljic delivery coefficient (newer than Gavrilovic coefficient) and is shown in Table 11.

**Table 11.** River sediment contribution for each river.

| River /$Q_t$ [m$^3$/year] | 1986-2006 | 1986-2001 | 2001-2006 |
|---|---|---|---|
| Stilaro | 72970 | 71364 | 75905 |
| Assi | 32947 | 32564 | 33707 |

By analysing the results reported in Table 11 we observe that the Stilaro river has a sediment contribution about 70,000 m$^3$/year and the Assi river has a sediment contribution about 30,000 m$^3$/year, with the variation of this parameter negligible

10   in both streams and in all intervals considered.



## 7 Quick and temporary solution

A storm in early December 2013 eroded much of the beach and has caused failure of parts of the dune and sliding of some archeological finds on the beach below (Fig. 17a). Following this storm, the Provincial Administration of Reggio Calabria in January 2014 built a barrier to protect the dune and this important artistic heritage. It was, however, a temporary

intervention. In fact, in early February 2014 another storm caused further damage to the beach and to the site (Fig. 17b, c).

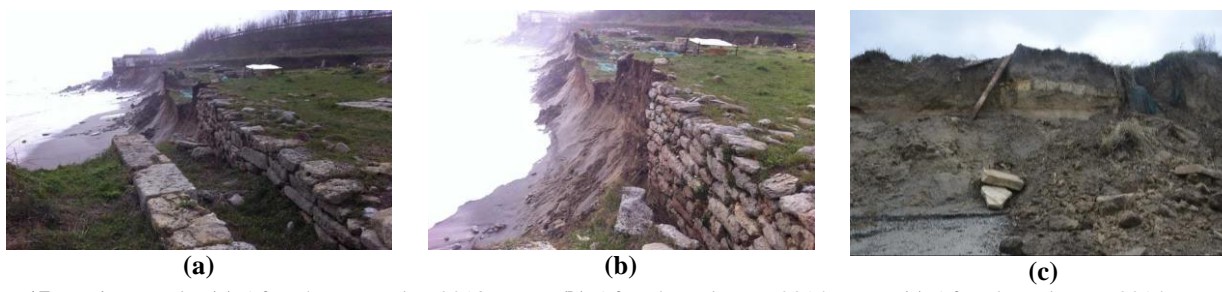

|       (a)       |       (b)       |       (c)       |

**Figure 17.** Doric Temple. **(a)** After the December 2013 storm. **(b)** After the February 2014 storm. **(c)** After the February 2014 storm.

In general, the direct observation of the effects of a storm on a coast improves the understanding of coastal dynamics of that locality. In this case, the analysis of the storm that occurred between in January 2014 to protect the archaeological site.

The characteristics of the storm were obtained by transposing the data recorded by the wave buoy of Crotone, by the geographical transposition model wave.

The storm recorded in Crotone between January 31 and February 3, 2014, had a direction of 135° N, reached a peak of significant height of 5.75m, a maximum wave height value of about 11 m, and a peak period of 10.25s while the peak of the average wind speed was 15 m/s (Fig. 18). The significant wave height values transposed do not differ much from those

recorded in Crotone, in particular the peak of significant wave height was 5.62m and the peak period was 10.10s.

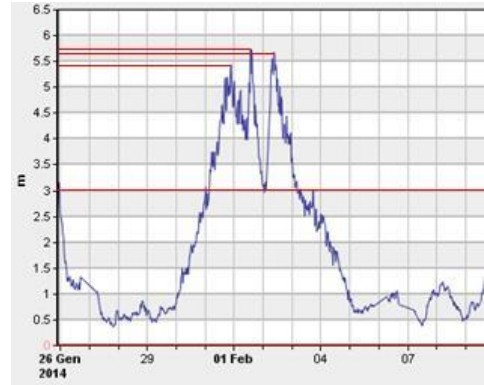

**Figure 18.** Storm recorded in Crotone between January 31 and February 3, 2014.

As mentioned above, following the storm of December 2013, the Provincial Administration of Reggio Calabria in January 2014 built a barrier to protect the dune and the important artistic heritage.



The barrier has a trapezoidal shape, is 3m in height, 6m in width and 30m long. The height between the ground level and the top of the barrier is 1.5m (Fig. 19). A sandy material was found to be deposited on the back of the barrier (Barbaro and Foti 2013), while at both ends an erosion of 4m of the dune was noted with respect to the axis of symmetry of the barrier, both due to the wave penetration.

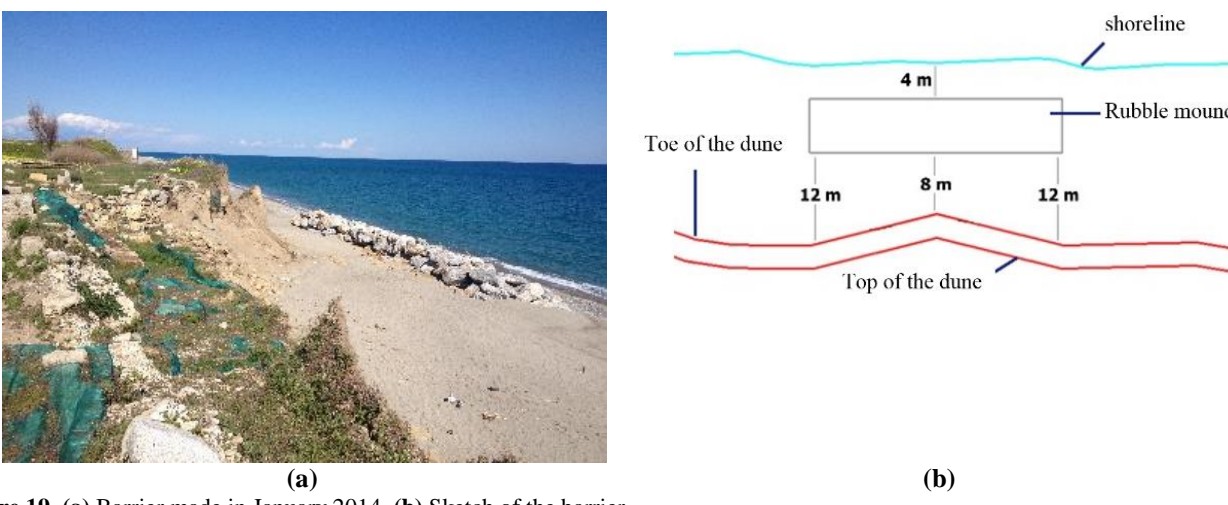

| (a) | (b) |

**Figure 19. (a)** Barrier made in January 2014. **(b)** Sketch of the barrier.

Before opting for the barrier, we evaluated other design solutions: in particular, the opportunity to build a wall was assessed, which in addition to protecting the site would also act to support the dune.

The walls, however, favour the appearance of wave reflection phenomena that lead to increased agitation wave (up to twice the height incident wave, due to the overlap of the incident and reflected wave fields) and the local depth of the seabed, in that the action of the breaking or reflected waves on it rapidly removes the sand at the foot of the wall, making it essential to the design of an adequate foundation to prevent it from slipping off. Therefore, this type of solution is not to be considered due to the complexity of the design problems and the huge expense that would result in their resolution.

The barrier in natural stones, however, limits the erosion phenomena due to its permeable structure which favours wave energy dissipation in its interior and not on the back of it. It is, also, a flexible structure that does not require special foundations. Finally, this type of work does not involve high manufacturing costs and the material used for its construction can be reused at the time of a definitive design solution.

A survey carried out in the month of February 2014 highlighted the barrier made in January 2014 has done its job in part because it withstood the storm of late January - early February 2014 but did not prevent the reoccurrence of further damage to the site and the dune (Fig. 20).



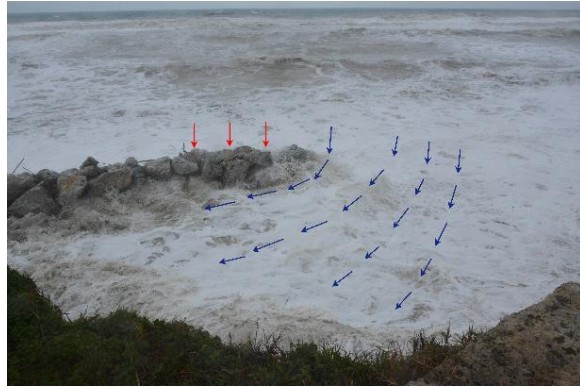

**Figure 20.** Wave penetration on the back of the barrier and circulation pattern.

## 8 Final solution

The final solution (Figs. 21 and 22), proposed by Calabria Basin Authority in 2015, aims to find a solution that ensures both adequate protection and the stabilization of the Monasterace beach whilst sustaining a stable and durable coast line, with an initial average progress of 40m compared to the present position of the beach. The proposed works involve a beach equal to 1,700 m, between the rivers Assi to the North and Stilaro to the South and they consist of:

- three T works of submerged natural boulders rooted to the shoreline with a semi-submerged groyne and barrier perpendicular to it, located at an height of submergence compared to the mean sea level equal to -0.50 m throughout its development. The semi-submerged groynes have a variable depth from 1.50m on the top to 0.50m on the bottom respect to the mean sea level. The groynes are 70m long. The submerged barrier is oriented parallel to the shoreline and has a length of 80m and is located at a variable depth from -3.50 m to -4.00m to the mean sea level;
- two sills, located at an average depth of -3.50 to -4.0m after the gaps between the T works of the length of 110m with a berm to -2.0m above the mean sea level, to contain the nourishment and dissipate wave power;
- stratified deposit of nourishment granular material along the 1,700m of coastline where defense works will be realized;
- dune system restoration.

This last point takes account of the fact that the most common coastal work is the beach-dune system as the sand eroded during a sea storm settles off forming bars of sediment parallel to the coast, causing a local decrease in the depth and frequent breaking waves, dissipating a significant amount of wave power. In addition, the dune ranges also represent a sand reservoir that feeds the process of formation of coastal bars: in fact, their destruction to make way for settlements and infrastructure is one of the reasons for the increase and diffusion of coastal erosion.

The drawing plan of intervention of coastal defense described above is shown in the following figures.





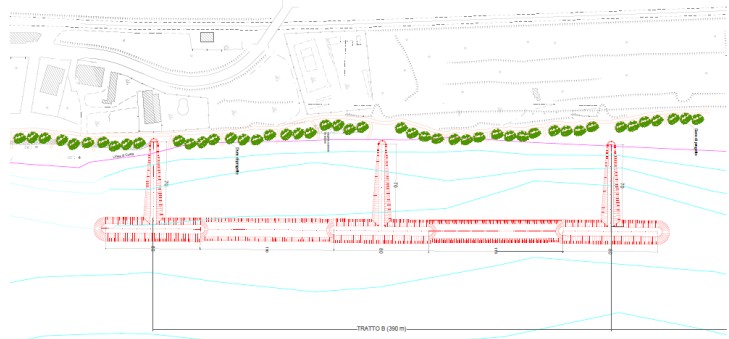

**Figure 21.** Final solution.

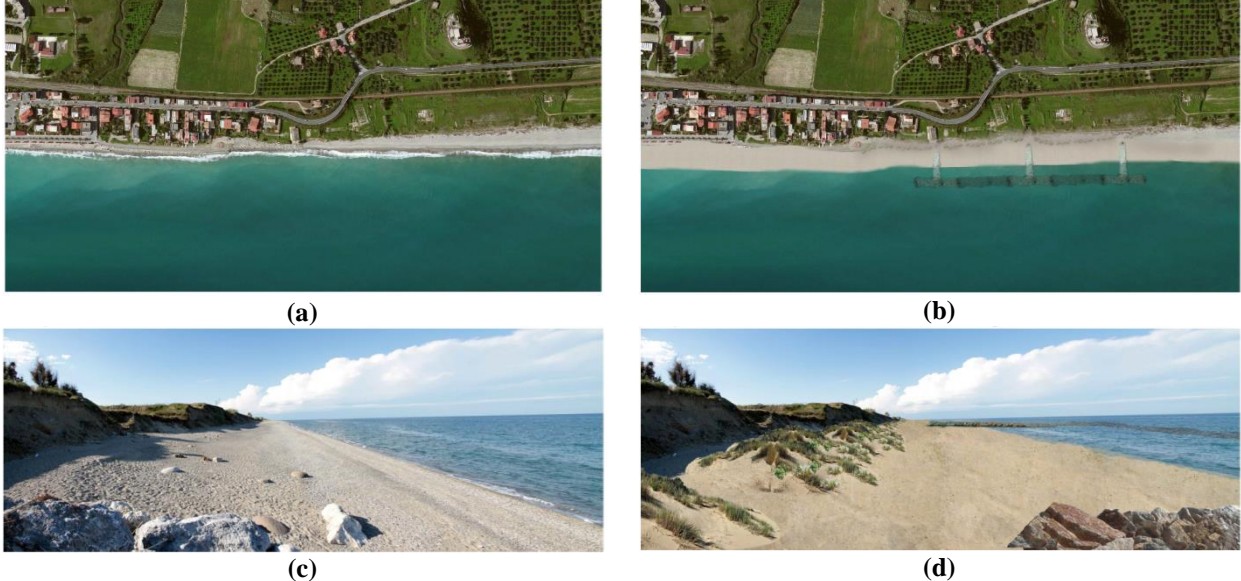

|  |  |
|---|---|
| **(a)** | **(b)** |
| **(c)** | **(d)** |

**Figure 22. (a)** Current state of the beach. **(b)** Design state of the beach. **(c)** Current state of the dune. **(d)** Design state of the dune.

**9 Conclusions**

5    The paper describes the erosive phenomena that affected the coast of Monasterace Marina in province of Reggio Calabria (Southern Italy), where the Kaulon archaeological site is located on a sand dune. All possible causes are investigated through the analysis of different types of data: bathymetric, cartographic, geological, geomorphological, sedimentological, climate and wave climate.  In particular, cartography data consists of aero photogrammetry by CASMEZ (1958), IGM (1958), aerial photos (1998 and 2008), provided by the Calabria Basin Authority (ABR), and satellite imagery provided by Google Earth

10  Pro. Wave time series starts on 10/01/1986 and finishes on 31/03/2006 and is provided by the Met Office database. To evaluate changes in river sediment transport, hydraulic structures, land cover data and weather and climate data, were analyzed, specifically the land cover data used related to years 2000 and 2006 (Corine Land Cover project), which is freely



available on the government agency website "Istituto Superiore per la Protezione e la Ricerca Ambientale (ISPRA)" while the weather and climate data are average rainfall registered in the stations near the area (Serra San Bruno, Monasterace Punta Stilo, Stilo Ferdinandea, Mongiana, Fabrizia, Stignano and Santa Caterina dello Ionio). This analysis was carried out for different historical periods over the last 50 years and has as its objective the evaluation of both the individual causes of

coastal erosion and the interactions that exist between them.

The site studied is located near the town of Monasterace Marina and between the mouth of the river Assi and Stilaro and is affected by the prevalent winds that blow from the South and South-Easterly and from North and North-Easterly directions. The most severe storm surges mainly from the South and South-Easterly direction, where the fetch is up to 700 km.

From the sedimentological and grain size distribution point of view, the beach sediments are composed of sand and light

gray gravels, with $D_{50}$ equal to 5 mm next to the isobaths of + 1.0m, $D_{50}$ = 0.7 mm next to the isobaths -3.0m and $D_{50}$ equal to 0.82 mm at the isobaths -7.0m.

The erosion in front of the archaeological site is evident, as it has developed over the last 50 years in two phases. Due to the lack of continuous monitoring of the coastline the following events happened in the time gaps. The first event happened between 1958 and 1985, during which phase more than the 30% of the original beach was eroded. The second time gap

phase was between 2001 and 2008, during which phase the erosion was more evident, and in fact was over 40%.

Between these two phases another one is evident, characterized by beach accretion, being between 1985 and 1998. Between 2008 and 2011 the coastline appears to have been in an equilibrium state.

From a wave climate point of view, the coast front Kaulon has been affected by frequent sea state from the northeast and southwest, with a maximum significant wave height about 1.4m and a maximum peak period about 5s, both from the sector

centred on the 100° N direction. Furthermore, the greatest part of the wave energy is between 40° and 200° N direction, with a maximum value centred on the 100° N direction.

Longshore sediment transport is directed by South from North direction and in the period 1986-2001 was slightly greater than that of the entire period while the interval 2001-2006 is significantly smaller. The run-up values are relatively low (they slightly exceed 1.5m only for the return period of 500 years) but they can still affect the stability of the dune.

Along the Assi and Stilaro rivers there are few check embankments and the river beds are formed principally by alluvial deposits.

The Stilaro river had a sediment contribution about 70,000 m³/year and the Assi river had a sediment contribution about 30,000 m³/year and the variation of this parameter is negligible in both streams and in all intervals considered.

By analysing the results described in the previous paragraphs it may be observed that the coastline is characterized by an

average annual longshore sediment transport, with direction South to North, equal to approximately 100,000 m³/ year, slightly higher in the range from 1986 to 2001 (about 110,000 m³/ year) and significantly lower in the range from 2001 to 2006 (about 60,000 m³/ year).

In addition, the average annual river sediment is about 70,000 m³/ year from Stilaro (located at the southern end of the area under consideration) and about 30,000 m³/ year from the river Assi (located at the northern end of this area) and this did not



change significantly in different time intervals. During this period, also, dams or other structures of significant size have not been built, and changes in land use destination are minor, so these factors do not change significantly the movement of transported fluvial sediments.

By crossing these results with the historical evolution of the coastline and the wave climate it can be observed, for the time

interval 1986-2001, the shoreline advanced while in the interval 2001-2006 it eroded. In the first interval, however, the wave was more intense than the second and the longshore sediment transport was higher than the river sediment transport produced by the Stilaro (which feeds the shoreline under consideration, following the trend in the longshore sediment transport directed from South to North). Instead, in the interval 2001-2006 the river sediment from the Stilaro river was higher than the average coastal transport.

It is, therefore, an unexpected result observed in comparison of the evolution of the coastline. Some possible explanations could be the river systems interventions and the arrangement of little slopes which were spatially extended and not recorded. These reduce the amount of eroded sediments or the dispersion beyond the closure depth, during sea storms and river floods, of small size sediments.

Finally, a temporary and a final solution are also described, to solve the erosive phenomena in the Monasterace Marina

coast, and to protect the Kaulon archaeological site.

## Acknowledgements

The authors would like to thank the Calabria Basin Authority for the material supplied and the constant support in the drafting of this paper.

Part of the work was carried out within the SIMONA Project (Systems and technologies for the monitoring of cultural areas

in underwater and terrestrial environment, POR Calabria 2007/2013).

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
