# Peer review of "Erosive phenomena in the Kaulon archaeological site: origins and remedies"

_Natural Hazards and Earth System Sciences, 2016_

## Referee Comment (RC1) · Anonymous Referee #1 · 7 Mar 2017

The manuscript entitled: "Erosive phenomena in the Kaulon archaeological site: origins and remedies" submitted by Barbaro et al. to Natural Hazards and Earth System Sciences provides an analysis of the shoreline changes occurring in the coast of Monasterace Marina (Calabria, Italy), proposes possible causes of erosion and describes the temporary and final solutions adopted to protect the Kaulon archaeological site that has been affected by beach and dune erosions during the winter 2013/2014 storms.

This manuscript is an extension of the article with a similar name already published [Barbaro, G., Foti, G., & Sicilia, C. L. (2016). Erosive Phenomena in the Proximity of Kaulon Archaeological Park: Origins and Remedies. Procedia-Social and Behavioral Sciences, 223, 714-719] that is not cited by the authors.

In my opinion this manuscript has to be rejected as it does not improve the results already published and it does not meet the scientific requirements to be published in Natural Hazards and Earth System Sciences.

The manuscript presented here is much longer than the already published article but the largest part of the extension does not help in understanding the problem and is out of scope of the Natural Hazards and Earth System Sciences journal. The main weaknesses are given as follows:

1. The causes of erosion still remain unclear (see for instance the penultimate paragraph of the conclusion) and further work is needed. A possible reason for the lack of correlation between the historical evolution of the shoreline and the wave climate (or the computed sediment transport balance that is one of the new part of the manuscript) is the 5 year wide time window where the climate data have been averaged (for the period 1986-2001 and 2001-2006). This limitation has been pointed out in the already published article, not in the present manuscript. Another limitation is that there is no shoreline data for the period 2001-2006. The erosion observed between 2001 and 2008 could have occurred during the period 2006-2008. Furthermore, the strong erosion events occurring in winter 2013-2014, following which interventions have been necessaries, are not included in the analysis.

2. Another new part of the manuscript is the description of the final solution (Section 8). However, similarly to the temporary solution (Section 7), only a brief description is given and the design is not rigorously justified. A strong weakness is that the design of these solutions is not related to the analysis previously performed to identify the causes of erosion (Section 5 and 6).

3. Some parts have been unnecessarily lengthened. The introduction is considerably lengthened but is mostly dedicated to generalities that are not related to the objectives of the manuscript and to introduce the coastal management strategies adopted in Calabria region that is out of scope of Natural Hazards and Earth System Sciences. A

large part of the site description (Section 3) is not necessary.
* * *

---

## Referee Comment (RC2) · Anonymous Referee #2 · 23 Mar 2017

GENERAL COMMENTS

The paper describes the erosive phenomena occurring along the coast of Monesterace Marina (Calabria, Italy) and investigates on possible causes of shoreline changes through the analysis of different types of data. In addition, a temporary and a final solution to protect the Kaulon archaeological site affected by beach and dune erosion is proposed. In the present form, the paper is redundant, verbous and lacks a consistent scientific analysis and it does not provide substantial and original results; there is a potentially very interesting available dataset, but the paper could do with a more extensive analysis by not including previously published material [e.g. Barbaro, G., Foti, G., & Sicilia, C. L. (2016). Erosive Phenomena in the Proximity of Kaulon Archaeological Park: Origins and Remedies. Procedia-Social and Behavioral Sciences, 223, 714-719]. As a consequence, the reviewer recommends reconsideration of the paper

following a major revision.

DETAILED COMMENTS

• Pag. 3, line 15: suggest to use "estimation" instead of "calculation". • Pag. 3, line 30: it seems redundant to consider an entire Section for listing the main aims of the paper. The objectives of the work can be briefly described in the introduction (line 18 to follow). • Pag. 4, line 12: suggest to substitute "Geographic" classification with "Geographical" classification. • Pag. 7 to pag. 11: the quality of Figure 5 to 9 has to be improved. • Pag. 10, line 3: are you referring to the mean peak wave period ? Please, explain. • Pag. 11, line 18: symbol 'h' has not been defined. • Pag. 14, line 15: please, define Tm as mean wave period. • Pag. 15, Eq. (16): please, explain the meaning of "per". • Pag. 15, Figure 11: the quality of Figure 11 has to be improved. • Pag. 16, line 3: "The run-up has been calculated using a probabilistic based on the ETS model...". Do you mean "The run-up has been calculated using a probabilistic approach based on the ETS model..." ? • Pag. 16, line 11: Please, define "X" as fixed treshold. Suggest to change the sentence "Return period of a run-up level higher than a fixed treshold is determined as...", with: "Return period of a run-up level higher than a fixed treshold, X, is determined as...". • Pag. 16, line 17: Please, define "D(X) as the mean persistence of the run-up above X". Suggest to change the sentence "The mean persistence of the run-up above the fixed treshold is given by..." with: "The mean persistence of the run-up, D(X), above X is given by...". • Pag. 17, Figure 12: Please, add axes labels. • Pag. 17, Figure 13: Please, add axes labels. • Pag. 17, line 8: "To evaluate changes in river sediment transport hydraulic structures, ...". Do you mean "To evaluate changes in river sediment transport in presence of hydraulic structures,.. " ? • Pag. 17, line 12: suggest to substitute "cards" with "charts". • Pag. 18, line 3: suggest to substitute "Land cover data" with "Digital mapping". • Pag. 20, line 9: "where h is the average yearly temperature". Please, consider that symbol 'h' has been previously adopted; see comment Pag. 11, line 18 above. • Pag. 20, line 13: X, the soil

protection coefficient, has been previously used for a fixed treshold; see comments Pag. 16 above. • Pag. 21, line 5: the Zemljic delivery coefficient has been adopted. Please, give more details. • Pag. 22, line 9: "In this case, the analysis of the storm that occurred between in January 2014 to protect the archaeological site". The sentence is not clear to the reader. • Pag. 24, line 8: suggest to substitute "works" by "structures". • An improvement in the description of the final solution (Section 8) and temporary solution (Section 7) is needed. The design of these solutions should take into account the analysis previously performed to characterize the causes of erosion. • A revision of the manuscript to verify English correct grammar and syntax is needed.

Please also note the supplement to this comment:
http://www.nat-hazards-earth-syst-sci-discuss.net/nhess-2016-399/nhess-2016-399-RC2-supplement.pdf